# Antibiotic Consumption at the Community Level: The Potential of a Single Health Policy Instrument to Assist Appropriate Use—Insights from Bulgaria

**DOI:** 10.3390/antibiotics14090888

**Published:** 2025-09-03

**Authors:** Desislava Vankova, Nadia Veleva, Petya Boncheva, Katerina Kondova, Zhaneta Radkova, Silviya Mihaylova

**Affiliations:** 1Faculty of Public Health, Medical University of Varna, 9002 Varna, Bulgaria; petya.boncheva@mu-varna.bg; 2Faculty of Pharmacy, Medical University of Pleven, 5800 Pleven, Bulgaria; nadia.veleva@mu-pleven.bg; 3Independent Researcher, 8000 Burgas, Bulgaria; katerinaa1@mail.bg; 4University Publishing Department, Medical University of Varna, 9002 Varna, Bulgaria; radkova@mu-varna.bg; 5Medical College, Medical University of Varna, 9002 Varna, Bulgaria; silviya.mihaylova@mu-varna.bg

**Keywords:** antimicrobial resistance, electronic prescribing, e-Rx policy, antibiotic stewardship, primary care, community antibiotic use, health policy implementation, Bulgaria, One Health

## Abstract

**Background:** Antimicrobial resistance (AMR) is accelerated by inappropriate antibiotic use in community settings. While most EU countries have achieved a statistically significant reduction in antibiotic consumption, Bulgaria has shown the opposite trend. The aim of this study is to investigate the implementation of mandatory electronic prescriptions (e-Rx) for antibiotics in Bulgarian primary care and to analyse community-level sales trends (2022–Q1 2025) in the context of related policy changes. **Methods:** The study applied a content analysis approach to publicly available policy documents and antibiotic sales and prescription data (IQVIA Bulgaria, NHIS). Participatory approaches facilitated the contextual interpretation of the data. The Health Policy Triangle framework guided the analysis of e-Rx implementation across four dimensions: content, context, process, and key actors involved in the e-Rx policy rollout. Trends in sales were assessed before and after the policy’s full enforcement in April 2024. **Results:** Sales data from IQVIA Bulgaria show a steady ≈10% decline in outpatient antibiotic sales from 2022 to 2024, with over 1.1 million fewer packages dispensed. Although the estimated annual and quarterly declines in community sales did not reach statistical significance in the short term, the consistent downward trajectory remains noteworthy. **Conclusions:** Mandatory e-Rx has shown early potential as a policy instrument to reduce antibiotic overuse in Bulgaria. It is expected to contribute to the reduction in AMR and to support the implementation of integrated national One Health policies.

## 1. Introduction

Antibiotics save lives. However, inappropriate and excessive use of antibiotics for human health are key determinants exacerbating antimicrobial resistance (AMR) [1]. According to the latest ESAC-Net data, the European Union (EU) is reporting a statistically significant decrease in the total consumption of antimicrobials (period 2013–2022) [2], which is a sustainable step towards the EU One Health strategy to manage AMR and to be a best-practice region [3]. However, Bulgaria deviates from overall EU trends, being the only Member State to exhibit a statistically significant increase in total antimicrobial consumption, and specifically antibiotic use at the community primary care level during the same period. This rise is largely attributable to non-biomedical determinants [1,4,5].

Despite national efforts, public awareness campaigns have demonstrated limited success in reducing antibiotic use. This underscored the pressing need for innovative interventions aligned with European Commission policy recommendations, particularly cross-border synchronisation of healthcare systems through proven ICT-enabled eHealth solutions, known as eHealth [6].

The first efforts toward digitalisation in Bulgaria’s healthcare system were outlined in the National Healthcare Strategy 2008–2013 [7]. In early 2012, the Integrated Health Information System (IHIS) was launched as part of the Base for Health Information System (BaHIS), but the project was discontinued. Subsequently, the Programme for the Development of Electronic Healthcare was initiated within the implementation plan of the National Healthcare Strategy 2020 [7]. Nevertheless, reforms in healthcare digitalisation had been postponed for years. The COVID-19 pandemic and the resulting social isolation, however, simultaneously exposed these shortcomings and accelerated the implementation of certain measures. In 2020, the National Health Information System (NHIS) was launched, initially providing patients only with COVID-19-related services such as vaccination records and certificates [8].

Building on these developments, this article examines a key subsequent step in Bulgaria’s digitalisation efforts—the introduction of mandatory electronic prescribing (e-Rx) for antibiotics. The academic ambition is to foster actors-centred and culturally sensitive research [9], highlighting and contextualising the e-Rx in Bulgaria as a timely health policy instrument, increasingly recognised for its potential to transform national prescribing practices and consumption behaviours, while addressing a critical gap in the European response to AMR.

The study relies on the theoretical framework of the classic Health Policy Triangle model by Gill Walt, which analyses policy interventions, as the e-Rx, through four criteria: historical, social and cultural context, the content and process of the e-Rx introduction, and the actors [10]. The actors include the key stakeholders at the primary healthcare level in Bulgaria—prescribers, community pharmacists, and patients or consumers, as well as the national health policy authorities as the Ministry of Health (MoH).

Historically, the country has struggled with antibiotic misuse, fuelled by a cultural context toward self-medication and lax access to prescription medications. According to EU Eurobarometer, a significant proportion of Bulgarians still hold misconceptions such as “antibiotics kill viruses” and are unaware that “unnecessary use of antibiotics renders them ineffective” [11]. Self-medication remained widespread— reported at 43.4% in 2014 [12] and 28.3% in 2022 [13], including the use of leftover antibiotics, taking antibiotics prescribed for someone else [12], and, in 26.9% of cases, a willingness to purchase antibiotics without a prescription [13]. This limited understanding has led to counterproductive behaviours, such as antibiotic overconsumption.

The specific social context is shaped by traditionally well-developed antibiotic stewardship system and by a highly centralized national healthcare system, based on mandatory social health insurance, which remains heavily oriented toward clinical care and continues to prioritise pharmaceutical and hospital expenditures over preventive services [14]. This imbalance has resulted in high out-of-pocket payments for primary care in Bulgaria, particularly for medicines. Notably, antibiotics in the primary healthcare sector are largely paid directly by the consumers or the patients. Importantly, this pharmaceutical demand for essential medications, such as antibiotics, demonstrates low price elasticity (PED < 1), meaning that inflation measures like price increases could not lead to substantial reductions in consumption. Moreover, based on data from the Bulgarian National Statistical Institute (NSI), general inflation levels remained relatively low—around 2–3% during the observation period.

With regard to AMR control, Bulgaria has established a National Reference Laboratory for Control and Monitoring of Antimicrobial Resistance [15], operating within the National Centre of Infectious and Parasitic Diseases. The laboratory is responsible for the detection and investigation of AMR and coordinates a national network of more than 150 microbiology laboratories—public, hospital-based, and private—covering all regions of the country and reporting antimicrobial consumption data to ESAC-Net. Reporting on AMR to the European Antimicrobial Resistance Surveillance Network (EARS-Net) is carried out directly by certified clinical microbiology laboratories on a voluntary basis. Further, the medical and pharmacy experts, including microbiologists and infectious disease epidemiologists, recognize AMR as an escalating public health threat in the country, driven by excessive antibiotic consumption [11,16,17]. However, AMR is not just specialists’ responsibility, and the outlined state-supported surveillance system predominantly covers hospital care, while information on antibiotic stewardship at the primary care level remains limited, warranting further research and policy attention.

The aim of this study is twofold: first, to trace the implementation process of mandatory e-Rx for antibiotics at the community or primary healthcare level in Bulgaria; second, to present follow-up analyses of trends in community-level antibiotic sales—used as a possible proxy for consumption behaviours—from 2022 through the first three months (Q1) of 2025, alongside a review of key health policy fluctuations associated with the introduction of e-Rx for antibiotics in primary care.

This research is both aligned with the World Health Organisation (WHO) Health in All Policies (HiAP) principles [18] and with the WHO One Health Priority Research Agenda for AMR, particularly the two research pillars—behavioural drivers and regulatory instruments [4]. Further, the analysis aligns with the EU’s objective of reducing human antibiotic consumption by 20% by 2030 [5]. By focusing on antibiotic use in primary care, this study also aligns with the WHO recommendations to monitor antibiotic consumption according to the AWaRe classification (Access, Watch, Reserve) [19]. At least 65% of total human antibiotic consumption should come from the Access group, which includes antibacterials effective against a broad range of commonly encountered susceptible pathogens and associated with a lower resistance potential [1]. In contrast, antibiotics in the Watch group carry a higher risk of resistance, while those in the Reserve group represent ‘last-resort’ treatment options [1,19]. The present study examines not only the overall level of antibiotic consumption but also its distribution according to the WHO AWaRe framework, thereby highlighting its broader social relevance. Given that AMR has cross-border implications, every community-level study on antibiotic use constitutes an essential component of the global AMR landscape.

## 2. Results

The Results section presents the e-Rx introduction in Bulgaria through the Health Policy Triangle framework focusing on the content and the implementation processes of the mandatory e-Rx for antibiotics.

Specifically, antibiotic sales data (annual, monthly and per AWaRe groups of antibiotics), before and after the implementation of e-Rx, are presented. The dataset covers only antibacterial drugs, and it does not include antivirals, antifungals, or antiparasitics. These sales figures are treated as possible real-time indicators of shifts in AMR-related behaviours, and specifically in antibiotic consumption. In this context, health behaviours at the community level are analysed as dynamic phenomena, varying over time, across different population cohorts, and across healthcare settings [20].

### 2.1. Content and Implementation Process of the Mandatory e-Rx for Antibiotic Use at the Community Level in Bulgaria

The health policy instrument under examination—mandatory e-Rx for antibiotics for systemic use at the community level in Bulgaria—is positioned within the broader context of healthcare digitalisation processes across the EU, which has formally started in 2008 [6].

As a component of NHIS, e-Rx as a prescription possibility was realised in Phase 2 of NHIS development and has been functioning since June 2021. It covers the dispensation of products with the so-called white prescriptions, and it is mandatory for medicinal products partially or fully reimbursed by the National Health Insurance Fund (NHIF). In June 2023, the fully electronic prescription of medicinal products dispensed with yellow or green prescriptions was introduced.

The initial national policy steps toward introducing e-Rx for antibiotics began earlier with the revision of Ordinance No. 4 (2009) on the Conditions and Procedure for Prescribing and Dispensing Medicinal Products. This revision provided more detailed guidance on the information required in prescriptions for prescription-only medicines—antibiotics included—and clarified pharmacists’ responsibilities in ensuring proper dispensing. A crucial amendment, introduced in Paragraph 70, mandated the use of electronic prescriptions specifically for antibacterial agents. Although this regulatory change was scheduled to come into force in November 2022, its implementation was delayed due to political instability and resistance from professional stakeholders.

Specifically, the first actual application of mandatory e-Rx for antibiotics, after pilot projects in selected regions, occurred in October 2023. This e-Rx policy was introduced in the post-COVID-19 context, at a time when digital health tools had gained broader cultural acceptance and increased support from stakeholders within the Bulgarian healthcare system. However, it faced significant opposition from stakeholders, citing issues such as time-consuming procedures and inadequate IT infrastructure. These challenges prompted a temporary moratorium on mandatory e-Rx, lasting from 19 December 2023 to 31 March 2024 (this was in fact the winter season, during which period physicians were allowed to continue issuing paper prescriptions alongside electronic ones). Ultimately, in April 2024, the e-Rx policy was fully and sustainably implemented. From that point forward, all primary care physicians were required to use digital platforms to prescribe antibiotics, ensuring that prescriptions were traceable, standardized, and centrally stored. The policy aimed to eliminate loopholes associated with manual prescribing, curb overprescription and overpurchase (as a single paper prescription can be used to obtain antibiotics from multiple pharmacies), and enhance the monitoring and governance of antimicrobial use.

All these processes have been regulated by the Ministry of Health, which is the authority responsible for enforcing the legal framework governing the porcesses of prescription and dispensing of antibiotics. Further, participatory approaches and key stakeholders’ interviews identified several structural and operational challenges during the early implementation phase of mandatory e-Rx for antibiotics in Bulgaria:Workflow inefficiencies—The prescription system requires multiple procedural steps to fulfil even routine prescriptions, leading to delays in service delivery and limiting pharmacists’ ability to provide comprehensive pharmaceutical care. This often results in patient dissatisfaction.Limited patient awareness—Patients are frequently unaware of the medication prescribed or its intended use. In most cases, they only present their national identification number at the pharmacy. Pharmacists are then expected to explain the therapy without access to diagnostic information or therapeutic rationale, as this data is not included in the electronic prescription.Unavailable or non-substitutable medications—Physicians occasionally prescribe products that are not currently available on the market. When generic substitution is not authorized, patients must revisit their prescribers to obtain an alternative prescription, creating additional burden on both sides.Partial fulfilment due to system gaps—Some prescription-only medicines are not listed in the NHIS, rendering full execution of certain prescriptions impossible.Product code inconsistencies—Discrepancies in NHIS product codes can block dispensing.Prescribing errors and inflexibility—Mistakes in dosage or formulation by prescribers cannot be corrected within the system. Even obvious errors necessitate a new prescription, as edits are not permitted.Outdated product listings—The NHIS includes many obsolete medicinal products that are no longer imported or distributed, complicating searches and potentially misleading prescribers or pharmacists.Complex search requirements—Retrieving an e-Rx requires at least two data points, typically including the date of issue. For patients with temporary identification numbers, it is often unclear which identifier was used, leading to time-consuming searches and service delays.System downtime risks—In cases of power outages, internet connectivity issues, or platform malfunctions, patients may be left without access to essential therapies.

In this context, data from the NHIS and community-level antibiotic sales are presented and analyzed in an effort to capture the trends in antibiotic consumption following the introduction process of the mandatory electronic prescription (e-Rx) system for antibiotics.

### 2.2. Electronic Prescriptions and Sales Data on Antibiotics at a Community Level—2022–2025

The digitalisation of the prescribing ultimately seeks to prevent the processing of incorrect, duplicate, expired, or fraudulent prescriptions.

Evidence from the NHIS illustrates the scale of implementation of the e-Rx, indicating that, as of January 2025, the total number of electronic prescriptions issued through NHIS has exceeded 7.125 million. Of these, over 83% (nearly 6 million) were for antibiotics, while the remainder were for diabetes medications. Data from the NHIS platform indicates that since 1 April 2024, when prescriptions for systemic-use antibacterial drugs became fully electronic, nearly 4.5 million antibiotic prescriptions have been issued. Of these, almost half allow substitutions with an equivalent medicinal product [21].

In addition to the NHIS prescribing statistics, sales data from IQVIA Bulgaria offer a broader perspective on antibiotic use and trends. This data is first summarized on an annual basis (see Figure 1), and then disaggregated by month to capture temporal patterns throughout 2022, 2023, and 2024, as well as the first quarter of 2025 (see Figure 2).

To better understand antibiotic use in relation to global stewardship frameworks, the data are subsequently stratified by AWaRe categories (see Figure 3). Monthly outpatient or community antibiotic sales as a linear trend in Bulgaria from January 2023 to March 2025 are visualized in Figure 4 to enhance clarity and comparative insight across the e-Rx introduction process (see Figure 4).

**Figure 1 antibiotics-14-00888-f001:**
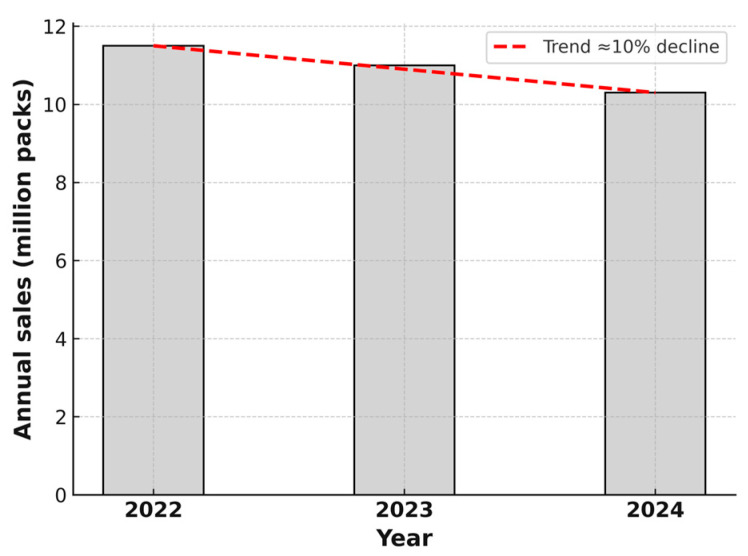
Total annual outpatient or community antibiotic sales in Bulgaria for the years 2022, 2023, and 2024. A decreasing trend is observed, with a cumulative reduction of approximately 10% in 2024 compared to 2022. Sales are expressed in millions of sales packs (individual commercial packages), based on national community pharmacy data. Data source: IQVIA Bulgaria. The figure is made by the authors.

Monthly trends from January 2023 through March 2025 provide further insights. The next chart presents monthly community (retail) antibiotic sales in Bulgaria over the period 2022 to March 2025 (Figure 2).

**Figure 2 antibiotics-14-00888-f002:**
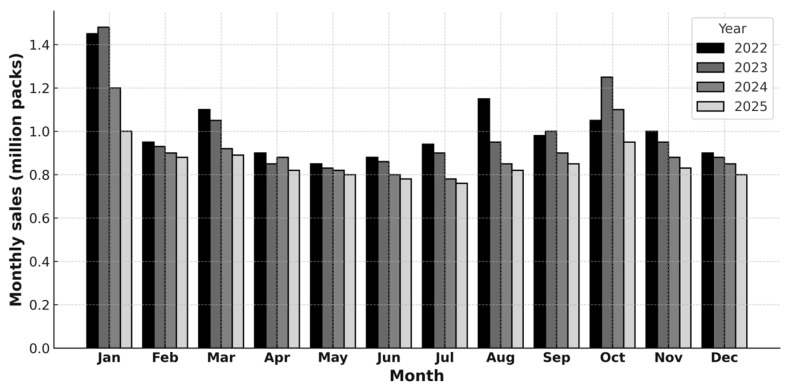
Monthly community antibiotic sales in Bulgaria by year—2022, 2023, 2024, March 2025. Monthly antibiotic sales exhibit a clear seasonality, with pronounced winter peaks and summer troughs. Sales are expressed in millions of sales packs (individual commercial packages), based on national community pharmacy data. Data source: IQVIA Bulgaria. The figure is made by the authors.

Figure 2 reveals distinct temporal dynamics in antibiotic sales, reflecting both seasonal fluctuations and the influence of regulatory changes. A clear seasonal pattern is observed, with sales typically peaking in January and reaching their lowest levels during the summer months. Overall sales declined modestly in 2023 compared with 2022, followed by a more pronounced decrease in 2024, most likely attributable to the implementation of new regulatory measures. Data for the first quarter of 2025 indicate that antibiotic sales remain at a lower level than in the corresponding months of previous years, despite the expected seasonal variations.

To better grasp the patterns of antibiotic use and the impact of the implementation of e-Rx, the data were stratified by AWaRe categories for the years 2022, 2023, and 2024. This stratification allows for enhanced comparative insight into shifts across the Access, Watch, and Reserve groups (Figure 3). Between 2023 and 2024, sales volumes declined across all AWaRe categories (Access, Watch, Reserve). The Access group experienced the largest absolute reduction − 74% (≈4.6 million packs), while Watch − 55% (≈2.4 million packs) and Reserve − 67% (≈0.8 million packs) also declined in parallel. These trends indicate a consistent downward trajectory across categories, with the most pronounced decline observed for Access antibiotics. Sales in the Watch and Reserve categories also decreased, although less steeply than in the Access group. The Watch group (e.g., macrolides and cephalosporins) and the Reserve group (e.g., fluoroquinolones, aminoglycosides, rifamycins)—which comprise antibiotics with higher resistance potential—experienced comparatively smaller reductions in sales volumes than the Access group, which includes essential first-line agents such as penicillins, tetracyclines, and trimethoprim combinations. The downward trajectories of these higher-risk categories are visually confirmed by the red trend lines in Figure 3.

**Figure 3 antibiotics-14-00888-f003:**
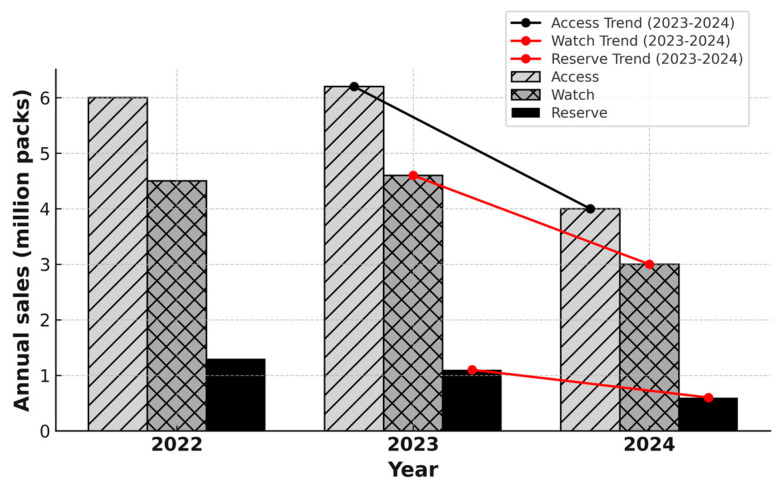
Annual outpatient antibiotic sales in Bulgaria stratified by WHO AWaRe categories (Access, Watch, Reserve) from 2022 to 2024. All categories showed a marked decline between 2023 and 2024: Access −74% (≈4.6 million packs), Watch −55% (≈2.4 million packs), and Reserve −67% (≈0.8 million packs).

Reserve groups: J01G fluoroquinolones, J01K aminoglycosides, J01M rifampicin and rifamycin, J01X other antibacterials. Watch groups: J01D cephalosporins and combinations, J01F macrolides and similar types, J01P other beta-lactam antibacterials excluding penicillins and cephalosporins. Access groups: J01A tetracyclines, J01B chloramphenicols, J01C broad-spectrum penicillins, J01E trimethoprim combinations, J01H medium/narrow-spectrum penicillins, S01A anti-infectives for eye, S01C ophthalmic anti-inflammatories with anti-infectives, D06A topical antibacterials.

According to the provided data by IQVIA Bulgaria, the community sales in packs per month following the introduction of the regulatory mandatory measure are presented in Figure 4—the e-Rx for antibiotics, which was first implemented in October 2023, temporarily suspended through a moratorium from December 2023 to March 2024, and reintroduced in a mandatory form as of 1 April 2024. The timeline visualization of the data in Figure 4 extends up to 31 March 2025, exactly one year after the final introduction (Figure 4).

During the initial launch of e-Rx in October 2023, there was no immediate observable impact on sales volume—likely due to a transitional implementation period or prescriber and pharmacy adaptation delays. The subsequent moratorium period, called “the transitional phase” (December 2023 to March 2024), coinciding with peak seasonal demand for antimicrobials, saw relatively stable sales. However, beginning in April 2024—when e-Rx became mandatory without exception—a clear and sustained decrease in monthly sales was observed. This downward trend coincides with a marked increase in the number of electronic prescriptions issued during the same period, according NHIS data [8].

**Figure 4 antibiotics-14-00888-f004:**
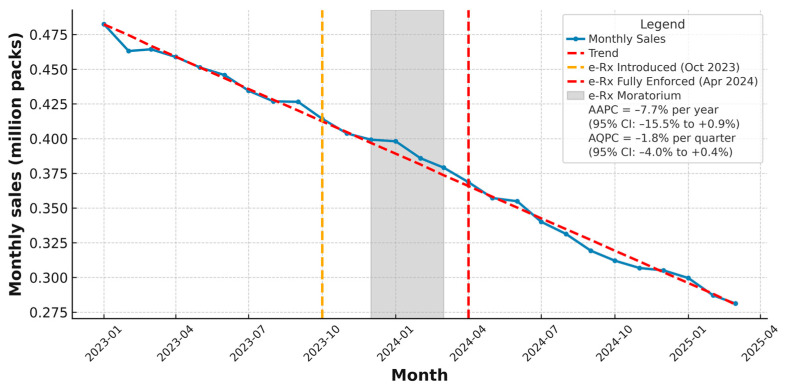
Monthly outpatient (community) antibiotic sales in Bulgaria from January 2023 to March 2025 (million packs). Key policy milestones are indicated: initial introduction of electronic prescriptions for antibiotics in October 2023 (orange dashed line), temporary moratorium between December 2023 and March 2024 (grey shaded area), and full enforcement in April 2024 (red dashed line). The blue line shows observed monthly sales, and the red dashed Trend line represents the overall fitted trend. Summary trend measures based on log-linear regression indicate an Average Annual Percent Change (AAPC) of −7.7% (95% CI: − 15.5% to +0.9%) and an Average Quarterly Percent Change (AQPC) of −1.8% (95% CI: −4.0% to +0.4%), reflecting a gradual overall decline in community antibiotic use during the study period. Sales are expressed in millions of sales packs (individual commercial packages), based on national community pharmacy data. Data source: IQVIA Bulgaria. The figure is made by the authors.

The summary trend measures—an Average Annual Percent Change (AAPC) of −7.7% (95% CI: −15.5% to +0.9%) and an Average Quarterly Percent Change (AQPC) of −1.8% (95% CI: −4.0% to +0.4%), reflecting a gradual overall decline in community antibiotic use during the study period. However, these findings indicate that while community antibiotic sales continued a gradual decline, the available monthly data do not demonstrate an additional, statistically significant drop or acceleration specifically attributable to the e-Rx policy in the short-term.

## 3. Discussion

The impact of the electronic prescription systems, including e-Rx for antibiotics, across the EU is indisputable. The quality of surveillance for appropriate antibiotic use varies between the EU countries, as does the enforcement of legislation preventing over-the-counter sales of antibiotics in community pharmacies [22]. Bulgaria with a population of ≈6,519,789 (last Census, 2021) is an EU Member State since 2017 and the digitalisation of the national healthcare follows the EU health policies. However, these reforms have been delayed for political and stakeholders’ resistance reasons. Bulgaria’s rollout of comprehensive e-health systems has lagged behind by decades compared to early adopters like Finland, which launched an electronic referral system as early as 1990, and the Netherlands and Germany, which had established key health informatics initiatives during the 1990s and early 2000s [23].

The Health Policy Triangle framework helped elucidate how the content, process, and actors intersected within the Bulgarian context. The insights from this specific national context show policy delays and cultural resistance.

Findings from population-based studies in Bulgaria highlight gaps in awareness and routine behaviours related to antibiotics among different actor groups. Although studies on adherence to good clinical practice standards that support rational antibiotic use are limited, several factors have been identified as contributing to the overprescription of antibiotics in the outpatient setting. While empirical antibiotic prescribing is clinically appropriate in many situations, such as community-acquired pneumonia or uncomplicated urinary tract infection, antibiotics are also frequently prescribed without prior antibiogram testing in cases where such urgency may not apply. This tendency is often driven by factors such as time pressure, cultural norms, fear of reputational or legal consequences, and action bias (the impulse to act in the patient’s perceived interest), which can contribute to inappropriate use and fuel AMR. Recent studies show that contributing factors include concerns about potential complications, limited time for consultations, financial incentives from pharmaceutical companies [24], prescribing ahead of weekends or public holidays, and long-term familiarity with patients, which may lead to prescribing without additional diagnostic testing [25,26]. For instance, in 2020, the NHIF reimbursed only 106,000 antibiograms, while community pharmacies dispensed over 12 million antibiotic packages—a trend that persisted in 2021 [27]. Moreover, there is documented evidence of antibiotic prescriptions being issued without in-person consultations, often following phone-based assessments [28]. However, these instances occurred prior to the mandatory implementation of e-Rx, with antibiotics recommended orally, without a formal written prescription. Current research is needed to reflect the prscription and dispensing practices after the introduction of the mandatory e-Rx for antibiotics.

The other important stakeholders are the community pharmacists and according to studies published in 2022, they feel confident when working with electronic prescriptions and they comply with the digitalisation of the Bulgarian healthcare [29,30,31]. Consequently, Bulgarian patients at the primary care level experience a more streamlined, yet stricter, access to antibiotics, and this process is supported by the community pharmacists. However, participatory approaches indicated that the early implementation of mandatory e-Rx for antibiotics in Bulgaria provided valuable insights into areas requiring refinement, including workflow efficiency, patient awareness, prescribing accuracy, and product listing updates. Addressing these challenging issues is expected to enhance access, optimize pharmaceutical care, and reduce burden for both patients and providers.

At the same time, according the NHIS, the first steps were successful and only about 2% of the prescriptions were cancelled. Training sessions were held for physicians and pharmacists, and the government provided technical support for integration with existing health IT systems. Challenges included technical delays, resistance from some practitioners, and data entry errors, all of which were gradually mitigated through targeted interventions [13]. However, analysis of NHIS data indicates that white e-Rx are primarily utilized for antibiotics and diabetes medications, owing to regulatory mandates requiring their electronic issuance. Other medicinal and non-medicinal products are typically prescribed using paper-based prescriptions or communicated verbally to patients, resulting in missing data within the NHIS. This highlights once again the critical importance of regulatory measures that mandate compliance with prescribing protocols and ensure the availability of reliable data. If healthcare professionals were to prescribe all medicinal and non-medicinal products electronically, which is already technically possible, this would generate more comprehensive information for stakeholders involved in shaping national health policies [8].

Harmonising the different perspectives, the current study analyses prove that the process of e-Rx for antibiotics implementation has been complex but ultimately successful, and actors showed a gradual adaptation. Importantly, the cultural readiness for digital tools in post-COVID times may have played a critical role in the acceptance and positive impact of this policy. Further, the stakeholders in Bulgaria have gradually accepted that the dispensing of antibiotics solely with e-Rx ensures transparency throughout the process, control over frequency and misuse without medical prescription, and effective tracking of the stock of medicinal products [32].

Sales data from IQVIA Bulgaria, used as a possible proxy for antibiotic consumption, reveal a marked decline in community-level antibiotic sales following the policy’s e-Rx full enforcement. Although the estimated 7.7% AAPC and 1.8% AQPC declines in community antibiotic sales did not reach statistical significance, the consistent downward trajectory remains noteworthy in the short-term. These findings suggest that overall prescribing and consuming behaviours may already be shifting in the desired direction, and that electronic prescribing could reinforce and sustain such trends as part of broader One Health stewardship strategies.

It is important to underline that the community sales of antibiotics, predominantly performed through out-of-pocket channels, could not be significantly influenced by the inflation in the country. Frist, because the inflation remained low for the last years. Second, pharmaceutical demand, especially for essential medications, such as antibiotics, demonstrates low price elasticity (PED < 1), meaning that moderate price increases do not lead to substantial reductions in consumption. These observations align with international evidence indicating that the demand for essential medicines, like the antibiotics, remains relatively stable despite economic fluctuations, as researched and documented by the Organisation for Economic Co-operation and Development (OECD) and other international organisations [33,34,35].

Monthly trend analysis in this study provided further insights, which include the overall seasonality, decline in sales, and a continued lower level of antibiotic sales. These patterns correspond with the policy implementation of e-Rx for antibiotics, initiated in October 2023, suspended temporarily during winter 2023–2024, and fully enforced in April 2024.

The analysis of annual antibiotic sales stratified by the WHO AWaRe classification demonstrates a consistent downward trajectory across all categories, with the Access group showing the most pronounced absolute and relative decline.

The reduction in Watch and Reserve antibiotics suggests that national antimicrobial stewardship measures are having some impact in limiting the use of antibacterials most strongly associated with resistance development. However, the comparatively steeper decline in Access antibiotics highlights the need for further interventions to promote the preferential prescribing of first-line agents. If such disproportionally smaller declines of Watch and Reserve persist, they could present a challenge for public health.

This observation is consistent with WHO guidelines, which recommend prioritizing the use of Access antibiotics while restricting Watch and Reserve categories to essential, targeted indications. Current sales trends therefore reflect suboptimal prescribing practices that call for strengthened professional education, improved clinical practice, stricter policy enforcement, and greater awareness among healthcare providers of the need to minimize inappropriate Watch and Reserve prescriptions in order to safeguard their effectiveness for life-saving use.

The study visualised the regulatory e-Rx-introduction policy milestones in a timeline which align with the steady decline of antibiotic community sales. This downward trend coincides with a marked increase in the number of electronic prescriptions issued during the same period, according NHIS data [8], suggesting that enhanced prescription monitoring and digital oversight may be contributing to more rational antibiotic use in the community setting. However, the statistical analyses indicate that while community antibiotic sales continued a gradual decline, the available monthly data do not demonstrate an additional, statistically significant drop or acceleration specifically attributable to the e-Rx policy in the short-term. Further long-term research is needed in this direction.

Public health studies indicate that individual behaviours and social circumstances together account for 60% of factors determining people’s health. Reducing antibiotic prescribing can happen with the help of harmonising health policy instruments like e-Rx with social norm feedback from the stewardship authorities. For example, doctors in the United Kingdom who had previously been identified as high prescribers of antibiotics reduced their prescriptions by 3.3% over a six-month period after receiving a letter from the country’s chief medical officer indicating that they prescribed more than most other medical practices in their local area [36,37]. Antimicrobial stewardship programmes at a national level should be designed to promote the most effective use of antimicrobials by limiting overuse, underuse, or misuse. Further research to support AMR-related national policies would hold significant social value.

In the longer term, e-Rx data can be linked with diagnostic and microbiological information to enable surveillance of antimicrobial resistance patterns and targeted stewardship interventions at a national level. The absence of an immediate statistically significant sales drop in the current study may reflect prescriber and patient adaptation periods, seasonal demand patterns, and the fact that behavioural change often requires sustained reinforcement beyond initial enforcement. Although the quantitative analysis did not identify a statistically significant acceleration in the downward trend of community antibiotic sales, the data and the overall analyses provide important feedback to the policymakers.

It is important to note that the observed decline in antibiotic sales during the e-Rx implementation period cannot be attributed solely to the policy itself. Other potential confounders and influences, such as concurrent public health interventions, fluctuations in infectious disease incidence, supply chain dynamics, distribution practices, and possible inappropriate underuse, may also have contributed to these trends. While our analysis highlights the temporal association with e-Rx introduction, further research incorporating comparative data (e.g., other therapeutic classes or regions without e-Rx) is needed to more precisely isolate its effect.

However, in Bulgaria there is yet no approved One Health National Action Plan to combat AMR nor an intersectoral One Health coordinating mechanism [38]. There was a draft National Programme for Rational Use of Antibiotics and Antibiotic Resistance Surveillance (2017–2021) [39,40], covering only the human health sector, but this was never adopted. A draft national action plan for animal health was developed separately. There was another draft covering the human, animal, environment, and food sectors, which was expected to be adopted in 2022, but the political changes again hindered the process [37]. Overall, there is limited attention to AMR due to other more pressing problems, e.g., constrained healthcare budget, high mortality rates in socially significant diseases like cardiovascular ones and cancer. Certainly, all these barriers should be addressed and there is political will for that. The new National Healthcare Strategy 2030 (NHS 2030) unites the fight against AMR and e-Rx, including as key policies both (1) to curb AMR through One Health approaches and the multisectoral collaboration, and (2) to further develop eHealth and digitalisation of the healthcare system [41], which covers the NHIS upgrade and the introduction of e-Rx as a key element of the digitalisation process [32]. Currently, a joint EU project on AMR titled EU-JAMRAI-2 [https://eu-jamrai.eu/] is gaining speed and the Bulgarian MoH is a partner, which could enforce the sustainable One Health strategies on AMR in the country.

## 4. Materials and Methods

This is a retrospective, observational epidemiological study combining quantitative sales data analysis with purposeful content analysis of public health documents (reports, strategies, etc.), and participatory approaches (our experts’ experiences). Variables evaluated include annual, monthly and quarterly antibiotic sales volumes before and after the e-Rx policy’s introduction, policy timelines, and stakeholder perceptions of prescribing and dispensing practices post e-Rx. Furthermore, data from the NHIS has been utilized.

Data sources: The primary source of data is provided by IQVIA Bulgaria, a global leader in healthcare data and analytics. As part of their commitment to social responsibility, IQVIA Bulgaria granted access to proprietary antibiotic sales data for the community (outpatient) sector, covering the period from 2022 to mid-2025. Antibiotic sales data were provided in sales packs, defined as individual commercial packages, as sold through the retail pharmacy channel. Each sales pack corresponds to one complete dispensing unit, irrespective of the dosage or number of tablets inside. The use of a uniform measurement unit, namely sales packs, ensures consistency in data reporting and allows for meaningful comparative analysis over time.

The IQVIA data reflects overall antibiotic sales from pharmacies, inclusive of both reimbursed and non-reimbursed (cash-based) purchases. This dataset does not directly include AMR surveillance data (e.g., microbial resistance patterns), but changes in antibiotic sales volumes are used as indirect indicators of behavioural and systemic shifts that relate to AMR management.

The IQVIA data capture the entire Bulgarian population, as it aggregates national-level sales data from retail pharmacies. It is not limited to insured patients or specific subgroups.

The analysis aggregates data on antibiotic consumption, sales and resistance data from several other open access sources [1,42,43,44,45]. The study builds on previous interdisciplinary research [46]. The presented study is evidence-based, following the idea that post-COVID health policies need to reflect global challenges as AMR and respect local cultural insights [47].

## 5. Conclusions

This study focuses on the policy relevance of the mandatory e-Rx for antibiotics. It examines the trends in community-level antibiotic sales in Bulgaria in the context of the gradual e-Rx implementation. The observed reduction in antibiotic use, including the WHO Watch and Reserve categories, coincides with key stages of the e-Rx policy rollout and suggests a possible association between digital prescribing and improved antibiotic stewardship.

Unlike traditional awareness campaigns, the introduction of e-Rx offers greater transparency, traceability, and accountability into the prescribing process. Through the lens of the Health Policy Triangle, our analysis highlights how policy content, implementation dynamics, and the broader socio-political context interact to influence health behaviours, concretely appropriate antibiotic consumption.

The observed average annual and quarterly declines in antibiotic sales, although not statistically significant, highlight a downward trend that is strengthen by the e-Rx introduction. While the analysis did not show a statistically significant acceleration in the decline of community antibiotic sales immediately after full e-Rx enforcement, electronic prescribing remains a valuable tool for antimicrobial stewardship. By enabling systematic monitoring, supporting adherence to prescribing guidelines, and providing data for targeted interventions, e-Rx can contribute to reducing inappropriate use in long term. Its full impact is likely to emerge through sustained application and integration into broader stewardship strategies.

At the same time, this experience underscores that raising awareness alone is insufficient to achieve sustainable change. Structural enablers such as digital tools must be embedded in wider stewardship frameworks to maximise their effect. While antibiotics save lives, they should not be treated as routine solutions or informal health guarantees. Public health initiatives must aim to institutionalise rational antibiotic use as a societal norm, supported by regulation, education, and feedback mechanisms.

Certainly, attributing reductions in antibiotic use solely to e-Rx would be premature. Structural enablers such as digital tools must operate within comprehensive stewardship frameworks to sustain impact. Further research is needed to evaluate the long-term effects of e-Rx, identify possible unintended consequences, and assess how complementary policies, such as clinician education, clinical decision support, and audits, can enhance its sustainability.

The Bulgarian case illustrates how digital health tools, aligned with HiAP principles, can contribute to addressing AMR and promoting evidence-based health governance. Finally, while Bulgaria’s e-Rx implementation marks a significant step forward, the absence of a fully adopted One Health National Action Plan and coordinating mechanism on AMR remains a key challenge. Integrating outpatient e-Rx systems into broader health governance structures, resistance surveillance, and intersectoral coordination is essential for the sustainable AMR control.

## 6. Strengths and Limitations of the Study

This study provides an integrated, interdisciplinary analysis of Bulgaria’s mandatory e-Rx policy for antibiotics, combining real-world sales data, stakeholder insights, and policy reviews within the robust framework of the Health Policy Triangle. A key strength of the study lies in its alignment with global AMR research priorities and its innovative use of national antibiotic sales data as a possible behavioural proxy at the community level. However, the study is limited by potential data inconsistencies across sources, the absence of Defined Daily Dose (DDD) and direct patient-level prescribing data, which may constrain causal inferences regarding behavioural change.

## Data Availability

The dataset provided, used, and analysed during the current study is publicly unavailable, supplied by IQVIA Bulgaria especially for the study.

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
