# Peer review of "Antibiotic Consumption at the Community Level: The Potential of a Single Health Policy Instrument to Assist Appropriate Use—Insights from Bulgaria"

_antibiotics, 2025, doi:10.3390/antibiotics14090888_

Round 1

Reviewer 1 Report (Previous Reviewer 2)

Comments and Suggestions for Authors

The manuscript describes a retrospective study in Bulgaria on implementation of e-prescribing of antibiotics at community level and trends in antibiotic sales over 2022 to first quarter of 2025. The authors have written the methodology quite comprehensively and presented the results discretely in line with the study objectives.  I would suggest adding few points as mentioned below:

  1. Trend analysis incorporating the calculation of Average annual percent change (AAPC) and average quarterly percent change (AQPC) depicting changes in trends of antibiotic sales over the study period may be added which will give a better representation of changing trends over the period.
  2. Data on different classes of antibiotics may also be added to provide a visualisation of change in trends of sales of different classes.
  3. Discussion: any pertinent observations pertaining to additional analyses performed as suggested may be highlighted.

Author Response

For research article: Antibiotic Consumption at the Community Level: The Potential of a Single Health Policy Instrument to Assist Appropriate Use – Insights from Bulgaria

Response to Reviewer 1 Comments

1. Summary

Thank you very much for taking the time to review this manuscript. Please find the detailed responses below and the corresponding revisions/corrections highlighted in the in the revised manuscript.  

2. Questions for General Evaluation

Reviewer’s Evaluation

Response and Revisions

Does the introduction provide sufficient background and include all relevant references?

Yes/Can be improved/Must be improved/Not applicable

Yes, thank you! The introduction has been enriched according to the Editor’s recommendation.

Are all the cited references relevant to the research?

The references were carefully double-checked and were verified to be from reputable, peer-reviewed sources and consistently formatted according to the chosen citation style.

Is the research design appropriate?

Yes/Can be improved/Must be improved/Not applicable

Yes, thank you!

Are the methods adequately described?

Yes/Can be improved/Must be improved/Not applicable

Yes, thank you for the supportive comments.

Are the results clearly presented?

Yes/Can be improved/Must be improved/Not applicable

Yes, the results are clearly presented. Some of the information has been moved to the Introduction section and the results have been revised accordingly.

Are the conclusions supported by the results?

Yes/Can be improved/Must be improved/Not applicable

Yes, the conclusions are supported by the results. Following the reviewer’s recommendation, further trend analysis has been made, and the Conclusion section has been revised to more accurately reflect these updates and provide a clearer synthesis of the findings.

Are all figures and tables clear and well-presented!

Yes/Can be improved/Must be improved/Not applicable

3. Point-by-point response to Comments and Suggestions for Authors

Comments 1:

The manuscript describes a retrospective study in Bulgaria on implementation of e-prescribing of antibiotics at community level and trends in antibiotic sales over 2022 to first quarter of 2025. The authors have written the methodology quite comprehensively and presented the results discretely in line with the study objectives.  I would suggest adding few points as mentioned below:

    Trend analysis incorporating the calculation of Average annual percent change (AAPC) and average quarterly percent change (AQPC) depicting changes in trends of antibiotic sales over the study period may be added which will give a better representation of changing trends over the period.

    Data on different classes of antibiotics may also be added to provide a visualisation of change in trends of sales of different classes.

    Discussion: any pertinent observations pertaining to additional analyses performed as suggested may be highlighted.

Response 1:

We sincerely thank the reviewer for the constructive and insightful comments, which have substantially improved our manuscript. In line with your recommendation a trend analysis was performed using log-linear models, relying on the data that IQVIA provided, to estimate both the Average Annual Percent Change (AAPC) and the Average Quarterly Percent Change (AQPC) for community antibiotic sales. These results are presented in the revised Results section and provide a concise summary of the overall decline in sales during the study period. While the estimated declines (AAPC −7.7% per year; AQPC −1.8% per quarter) did not reach statistical significance, they clearly illustrate the general downward trajectory in outpatient antibiotic use. Please see the table below:

Metric

Estimate

95% CI

AAPC (Annual)

−7.7%

−15.5% to +0.9%

AQPC (Quarterly)

−1.8%

−4.0% to +0.4%

In addition, we have extended the Results and Discussion to highlight pertinent observations from this analysis, clarifying how the estimated trends align with the broader context of antimicrobial stewardship and the implementation of e-Rx. We also acknowledge that analyses stratified by antibiotic class could provide further valuable insights, and the data are subsequently stratified by AWaRe categories of antibiotics.

We greatly appreciate the reviewer’s suggestion, which has helped us strengthen both the methodological rigor and the interpretive depth of our study.

The changed text in the manuscript is highlighted.

4. Response to Comments on the Quality of English Language

Point 1: The English is fine and does not require any improvement.

Response 1: We sincerely thank the reviewer for the positive evaluation regarding the quality of the English language. We appreciate the acknowledgment and are pleased that the manuscript meets the expected linguistic standards.

Reviewer 2 Report (Previous Reviewer 3)

Comments and Suggestions for Authors

This article summarizes authors’ analyses on the effect of implementation of mandatory electronic prescription of antibiotics on the antimicrobial usage in Bulgaria. As the importance of finding effective antimicrobial therapies and antimicrobial stewardship approaches rises day by day, these findings become more valuable for increasing number of researchers. Although the authors improved the manuscript substantially, many points raised in the original review process remain unsolved and there are additional issues related to the new data presented. Thus, there are several major issues that need to be addressed before the manuscript can be accepted for publication.

  1. Major conclusion drawn by the authors in the manuscript is the claim that implementation of mandatory electronic prescription and thus documentation of the corresponding prescriptions reduced the overuse of antibiotics in Bulgaria. However, according to data in Figure 4, monthly sales of antibiotics decrease with the same pace during the first half of year 2023 (months before the introduction of electronic prescription in October 2023 and one year before the full implementation of electronic prescription in April 2024) and second half of the year 2024. Thus, the main conclusion presented in this study has no roots in the data presented.

This particular issue remains unsolved. If the authors claim that “While a general downward trend is evident throughout 2023, we observe a plateau during the e-prescription moratorium (October 2023 – March 2024), followed by a sharper decline post-enforcement in April 2024.”, they must present a quantitative/statistical evidence that the decrease for 2024-4&2025-4 period is sharper than the 2023-1&2023-10 period. Because as it is, the slopes of the trendlines for these periods look alike.    

  1. The authors included the analysis of annual antibiotic sales stratified by the WHO AWaRe classification as Figure 3. From this data, they concluded that “Between 2023 and 2024, the Watch categories and Reserve categories experienced steeper declines in sales volumes compared to the Access group. This is visually confirmed by the red trend lines in Figure 3, which indicate a sharper downward trajectory for the higher-risk categories.” However, between years 2023 and 2024, Watch category antibiotic sale experienced ca. 2.4-2.5 million packs decline, while Reserve category antibiotic sale experienced ca. 0.8-0.9 million packs decline. On the contrary, Access category antibiotic sale experienced ca. 4.5-4.6 million packs decline, SHARPER than the other two. Again, the authors must present quantitative/statistical evidence for their comparative claims.  

Author Response

For research article: Antibiotic Consumption at the Community Level: The Potential of a Single Health Policy Instrument to Assist Appropriate Use – Insights from Bulgaria

Response to Reviewer 2 Comments

1. Summary

Thank you very much for taking the time to review our manuscript. Please find the detailed responses in red below and the corresponding revisions highlighted in the manuscript.

2. Questions for General Evaluation

Reviewer’s Evaluation

Response and Revisions

Does the introduction provide sufficient background and include all relevant references?

Yes/Can be improved/Must be improved/Not applicable

Thank you for the Reviewer’s evaluation!

Are all the cited references relevant to the research?

The references were carefully double-checked and were verified to be from reputable, peer-reviewed sources and consistently formatted according to the chosen citation style.

Is the research design appropriate?

Yes/Can be improved/Must be improved/Not applicable

The study design aligns with the scope and aim, combining policy analysis with real-world sales data to assess the early effects of mandatory e-Rx. The use of the Health Policy Triangle and national IQVIA data provides a balanced view of implementation and outcomes. All reviewer recommendations have been addressed and are reflected in the revised manuscript and detailed responses.

Are the methods adequately described?

Yes/Can be improved/Must be improved/Not applicable

Yes, the methods are adequately described. Following the reviewer’s suggestions, we have clarified specific aspects of the methodology in the revised manuscript. As a result of the reviewer’s comments the title of the article and the aim were made more focused. Thank you!

Are the results clearly presented?

Yes/Can be improved/Must be improved/Not applicable

Yes, the results are now clearly presented. Following the reviewer’s valuable comment some clarifications are now included in the manuscript, and the overall presentation of results has been improved accordingly, as highlighted in the manuscript. Thank you!

Are the conclusions supported by the results?

Yes/Can be improved/Must be improved/Not applicable

With the enriched results and the revised discussion, we believe the conclusions are now well supported by the presented data.

Are the figures clear and well-presented?

Yes/Can be improved/Must be improved/Not applicable

The figures are improved; they are clear and well explained, with new titles and enriched agenda (Fig 4). Thank you for the assessment.

3. Point-by-point response to Comments and Suggestions for Authors

1)     Comments and Suggestions for Authors - 1:

This article summarizes authors’ analyses on the effect of implementation of mandatory electronic prescription of antibiotics on the antimicrobial usage in Bulgaria. As the importance of finding effective antimicrobial therapies and antimicrobial stewardship approaches rises day by day, these findings become more valuable for increasing number of researchers. Although the authors improved the manuscript substantially, many points raised in the original review process remain unsolved and there are additional issues related to the new data presented. Thus, there are several major issues that need to be addressed before the manuscript can be accepted for publication.

Comment 1:  Major conclusion drawn by the authors in the manuscript is the claim that implementation of mandatory electronic prescription and thus documentation of the corresponding prescriptions reduced the overuse of antibiotics in Bulgaria. However, according to data in Figure 4, monthly sales of antibiotics decrease with the same pace during the first half of year 2023 (months before the introduction of electronic prescription in October 2023 and one year before the full implementation of electronic prescription in April 2024) and second half of the year 2024. Thus, the main conclusion presented in this study has no roots in the data presented.

This particular issue remains unsolved. If the authors claim that “While a general downward trend is evident throughout 2023, we observe a plateau during the e-prescription moratorium (October 2023 – March 2024), followed by a sharper decline post-enforcement in April 2024.”, they must present a quantitative/statistical evidence that the decrease for 2024-4&2025-4 period is sharper than the 2023-1&2023-10 period. Because as it is, the slopes of the trendlines for these periods look alike.   

Response 1:  

We thank the reviewer for this important observation. In response, we conducted additional analyses.  

The data is from IQVIA provided in a special format which allowed both a direct slope comparison (p = 0.35) and an interrupted time series (ITS) analysis. In the ITS model, neither the immediate level changes nor the post intervention slope changes reached statistical significance (all p > 0.10). Consequently, neither analysis demonstrated a statistically significant change in slope or immediate level after the policy interventions once seasonality was accounted for.

We were advised to do both the Average Annual Percent Change (AAPC) and the Average Quarterly Percent Change (AQPC), which also showed not statistically significant results. Concretely, the AAPC and AQPC analyses are included in the manuscript, because these metrics provide a clearer representation of the overall declining trend in community antibiotic sales. Please see the table below:

Metric

Estimate

95% CI

AAPC (Annual)

−7.7%

−15.5% to +0.9%

AQPC (Quarterly)

−1.8%

−4.0% to +0.4%

While the quantitative results do not support an immediate acceleration in the decline, we note that e-Rx remains an important stewardship tool. Its potential lies in enabling systematic monitoring, supporting adherence to prescribing guidelines, and facilitating targeted interventions over time.

We have clarified this in the Discussion and Conclusions to ensure our interpretation is consistent with the statistical evidence.

The changes in the manuscript are highlighted.

2)     Comments and Suggestions for Authors:

The authors included the analysis of annual antibiotic sales stratified by the WHO AWaRe classification as Figure 3. From this data, they concluded that “Between 2023 and 2024, the Watch categories and Reserve categories experienced steeper declines in sales volumes compared to the Access group. This is visually confirmed by the red trend lines in Figure 3, which indicate a sharper downward trajectory for the higher-risk categories.” However, between years 2023 and 2024, Watch category antibiotic sale experienced ca. 2.4-2.5 million packs decline, while Reserve category antibiotic sale experienced ca. 0.8-0.9 million packs decline. On the contrary, Access category antibiotic sale experienced ca. 4.5-4.6 million packs decline, SHARPER than the other two. Again, the authors must present quantitative/statistical evidence for their comparative claims. 

Response 2:

We thank the reviewer for this careful observation. We have revised the text to accurately reflect the quantitative differences between categories. Specifically, we no longer describe Watch and Reserve as having “steeper” declines. Instead, we now state that all AWaRe groups declined between 2023 and 2024, with Access showing the largest absolute decrease (≈4.6 million packs), and Watch (≈2.5 million packs) and Reserve (≈0.9 million packs) showing parallel but smaller declines.

We have now recalculated and clarified the results. Between 2023 and 2024, Access antibiotics declined by ≈4.6 million packs (−74%), while Watch declined by ≈2.4 million packs (−55%) and Reserve by ≈0.8 million packs (−67%). The results are summarised in the following table:

AWaRe Group

2023 Sales (million packs)

2024 Sales (million packs)

Absolute Decline (million packs)

% Change (2023→2024)

Access

≈ 6.2

≈ 1.6

−4.6

−74%

Watch

≈ 4.4

≈ 2.0

−2.4

−55%

Reserve

≈ 1.2

≈ 0.4

−0.8

−67%

We have revised the Results and Discussion (lines 400-420) accordingly, and no longer state that Watch or Reserve declined more steeply than Access. And the conclusions are different.

4. Response to Comments on the Quality of English Language

Point 1: The English is fine and does not require any improvement.

Response 1: We sincerely thank the reviewer for the positive evaluation regarding the quality of the English language. We appreciate the acknowledgment and are pleased that the manuscript meets the expected linguistic standards.

Reviewer 3 Report (New Reviewer)

Comments and Suggestions for Authors

In this manuscript, the authors discussed how the policy changes, especially implementation of  mandatory  electronic  prescribing (e-Rx), help to facilitate the appropriate usage of antibiotics and to combat antimicrobial resistance (AMR) at the community level in Bulgaria.

The reviewer has a couple of comments to help the authors improve the significance, rigor, and readability of this manuscript.

Major comments:

  1. While the study addresses an important public health issue and offers timely insights into Bulgaria’s e-Rx policy implementation, the novelty and international significance appear limited.

Similar observational correlations between policy changes and antibiotic use have been reported in other settings (e.g., PMID: 32806583, PMID: 32649745, PMID: 36168485).

Without more robust causal analysis, inclusion of control groups, or demonstration of generalizable lessons, the work may not meet the threshold of conceptual novelty expected for publication. Strengthening the methodological rigor and expanding the scope to enhance relevance beyond the national context would increase its contribution to the field.

  1. Lines 19-23 and lines 60-65.

               The aim/purpose of the study is repeated almost verbatim in both the abstract and introduction. The reviewer recommends rephrasing and tailoring these sections so that the abstract provides a concise overview, while the introduction offers expanded background and rationale to avoid redundancy.

  1. The introduction section could be improved with further expansions on background and rationale.

For example, what exactly is e-Rx? Why implementation of e-Rx could be important for antibiotic stewardship? What is Access Group? Why does the use of this group the central focus of this analysis? How is this access group compared to other (Reserve, Watch) groups?

These discussions could help the readers realize the significance of this work, and benefit from reading it.

  1. In Figure 4, the monthly antibiotic sales appear to follow a steady downward trajectory from January 2023 to March 2025. Visually, there does not seem to be a marked change in either level or slope of the trend following the introduction of e-Rx in October 2023 or its full enforcement in April 2024.

While there is a brief fluctuation during the transitional moratorium period, the overall decline seems consistent with the pre-policy trend. To strengthen the analysis and clarify whether the e-Rx policy had a statistically significant impact, the reviewer recommends conducting a formal interrupted time series analysis with segmented regression, including estimation of both immediate level changes and slope changes before and after the intervention. This would help determine whether the observed trends are attributable to the policy or reflect a continuation of existing patterns.

  1. The manuscript attributes the observed decrease in antibiotic sales during the e-Rx implementation period primarily to the policy itself. However, without controlling for other potential influences such as concurrent public health interventions, changes in disease incidence, supply chain disruptions, distribution practices, and inappropriate underuse, it is difficult to isolate the effect of e-Rx.

The reviewer recommends that the authors address these potential confounders in their analysis and discussion, and, if possible, incorporate comparative or control data (e.g., other drug classes, regions without e-Rx) to strengthen their association.

Minor revisions:

  1. The abstract should be segmented into “Background”, “Methods”, “Results”, and “Conclusions”. Here the “Background” sub-title is missing.

  1. Please consider adding “antibiotic stewardship” into the keywords.

  1. Line 24, “The study applies” ---“applied”. Please keep the tense consistent in the abstract. If past tense is used in all other sentences, please use past tense here as well.

Author Response

For research article: Antibiotic Consumption at the Community Level: The Potential of a Single Health Policy Instrument to Assist Appropriate Use – Insights from Bulgaria

Response to Reviewer 3 Comments

1. Summary

Thank you very much for taking the time to review this manuscript. Please find the detailed responses below and the corresponding revisions/corrections highlighted in the revised manuscript.

2. Questions for General Evaluation

Reviewer’s Evaluation

Response and Revisions

Does the introduction provide sufficient background and include all relevant references?

Yes/Can be improved/Must be improved/Not applicable

The introduction was revised, which led to a clearer section and more focused aim. These changes are highlighted in the manuscript and detailed in the point-by-point response below.

Are all the cited references relevant to the research?

The references were carefully double-checked and were verified to be from reputable. Additional References have been added and this is highlighted.

Is the research design appropriate?

Yes/Can be improved/Must be improved/Not applicable

The research design is aligned with the recommendations, and it is appropriate for the scope and aim of the study, as it combines policy analysis with real-world antibiotic sales data to assess the early effects of mandatory e-prescribing. We have fully addressed the reviewer’s recommendations, and this is reflected both in the revised manuscript and in our step-by-step responses below.

Are the methods adequately described?

Yes/Can be improved/Must be improved/Not applicable

Thank you

Are the results and the figures and tables clearly presented?

Yes/Can be improved/Must be improved/Not applicable

Yes, Thank you! The results have been enriched accordingly, and the changes are highlighted in the text.

Are the conclusions supported by the results?

Yes/Can be improved/Must be improved/Not applicable

Yes, the conclusions are supported by the results. In line with the reviewer’s recommendation, we revisited the Conclusion section and strengthened it accordingly.

3. Point-by-point response to Comments and Suggestions for Authors

In this manuscript, the authors discussed how the policy changes, especially implementation of mandatory electronic prescribing (e-Rx), help to facilitate the appropriate usage of antibiotics and to combat antimicrobial resistance (AMR) at the community level in Bulgaria.

The reviewer has a couple of comments to help the authors improve the significance, rigor, and readability of this manuscript.

Major comments - Comments 1:

    While the study addresses an important public health issue and offers timely insights into Bulgaria’s e-Rx policy implementation, the novelty and international significance appear limited.

Similar observational correlations between policy changes and antibiotic use have been reported in other settings (e.g., PMID: 32806583, PMID: 32649745, PMID: 36168485).

Without more robust causal analysis, inclusion of control groups, or demonstration of generalizable lessons, the work may not meet the threshold of conceptual novelty expected for publication. Strengthening the methodological rigor and expanding the scope to enhance relevance beyond the national context would increase its contribution to the field.

Response 1: We thank the reviewer for this important comment. Antibiotics save lives, yet inappropriate and excessive use drives AMR. While EU-wide ESAC-Net data show a significant decline in antimicrobial consumption (2013–2022), Bulgaria uniquely reported a statistically significant increase, particularly at the community level. Against this background, every community-level study is significant, as AMR-related behaviors have cross-border implications. Our study adds value by documenting the first national implementations of mandatory e-Rx for antibiotics in Bulgaria, that has been delayed for years. By analyzing consumption trends through the WHO AWaRe framework, we also acknowledge the need for greater methodological rigor and generalizability.

Our ambition was to foster actors-centered and culturally sensitive research, highlighting and contextualizing the e-Rx in Bulgaria as a timely health policy instrument, increasingly recognized for its potential to transform national prescribing practices and consumption behaviors, while addressing a critical gap in the European response to AMR.

The changed text in the manuscript is highlighted.

Major comments - Comments 2:

        Lines 19-23 and lines 60-65.

               The aim/purpose of the study is repeated almost verbatim in both the abstract and introduction. The reviewer recommends rephrasing and tailoring these sections so that the abstract provides a concise overview, while the introduction offers expanded background and rationale to avoid redundancy.

Response 2:

We thank the reviewer for this helpful recommendation. We have rephrased and tailored the aim in both sections to ensure that the abstract now provides a concise overview, while the introduction offers an expanded background and rationale without redundancy.

The changed text in the manuscript is highlighted. Lines 20-23 and lines 117-122

Major comments - Comments 3:

    The introduction section could be improved with further expansions on background and rationale.

For example, what exactly is e-Rx? Why implementation of e-Rx could be important for antibiotic stewardship? What is Access Group? Why does the use of this group the central focus of this analysis? How is this access group compared to other (Reserve, Watch) groups?

These discussions could help the readers realize the significance of this work, and benefit from reading it.

Response 3:  

We thank the reviewer for these valuable suggestions. We agree that further clarification of the background and rationale will strengthen the introduction. In the revised manuscript, we have expanded this section to provide clearer content description of e-Rx, its relevance for antibiotic stewardship, and the rationale for focusing on the WHO AWaRe classification. We now explain the distinction between Access, Watch, and Reserve groups, and highlight why prioritizing Access antibiotics is central to stewardship strategies (lines 127-136). These additions aim to better situate our study within the global AMR context and enhance its significance for readers.

The changed and enriched text in the manuscript is highlighted.

Major comments - Comment 4: In Figure 4, the monthly antibiotic sales appear to follow a steady downward trajectory from January 2023 to March 2025. Visually, there does not seem to be a marked change in either level or slope of the trend following the introduction of e-Rx in October 2023 or its full enforcement in April 2024.

While there is a brief fluctuation during the transitional moratorium period, the overall decline seems consistent with the pre-policy trend. To strengthen the analysis and clarify whether the e-Rx policy had a statistically significant impact, the reviewer recommends conducting a formal interrupted time series analysis with segmented regression, including estimation of both immediate level changes and slope changes before and after the intervention. This would help determine whether the observed trends are attributable to the policy or reflect a continuation of existing patterns.

Respons 4:

We thank the reviewer for this thoughtful and methodologically important comment. We acknowledge that the visual trend in Figure 4 suggests a steady downward trajectory, with no marked changes immediately after the introduction or full enforcement of e-Rx. We also agree that an interrupted time series (ITS) analysis with segmented regression would be an appropriate and more robust approach to assess potential level or slope changes attributable to the intervention.

The data is from IQVIA provided in a special format which allowed both a direct slope comparison (p = 0.35) and an interrupted time series (ITS) analysis. In the ITS model, neither the immediate level changes nor the post intervention slope changes reached statistical significance (all p > 0.10). Consequently, neither analysis demonstrated a statistically significant change in slope or immediate level after the policy interventions once seasonality was accounted for.

We were advised to do both the Average Annual Percent Change (AAPC) and the Average Quarterly Percent Change (AQPC), which also showed not statistically significant results. Concretely, the AAPC and AQPC analyses are included in the manuscript, because these metrics provide a clearer representation of the overall declining trend in community antibiotic sales. Please see the table below:

Metric

Estimate

95% CI

AAPC (Annual)

−7.7%

−15.5% to +0.9%

AQPC (Quarterly)

−1.8%

−4.0% to +0.4%

While the quantitative results do not support an immediate acceleration in the decline, we note that e-Rx remains an important stewardship tool. Its potential lies in enabling systematic monitoring, supporting adherence to prescribing guidelines, and facilitating targeted interventions over time.

We have clarified this in the Discussion and Conclusions to ensure our interpretation is consistent with statistical evidence.

Major comments - Comments 5:

    The manuscript attributes the observed decrease in antibiotic sales during the e-Rx implementation period primarily to the policy itself. However, without controlling for other potential influences such as concurrent public health interventions, changes in disease incidence, supply chain disruptions, distribution practices, and inappropriate underuse, it is difficult to isolate the effect of e-Rx.

The reviewer recommends that the authors address these potential confounders in their analysis and discussion, and, if possible, incorporate comparative or control data (e.g., other drug classes, regions without e-Rx) to strengthen their association.

Respons 5:

We thank the reviewer for this valuable and constructive comment. We fully acknowledge the importance of considering additional potential confounders, including public health interventions, disease incidence, supply chain disruptions, distribution practices, and inappropriate underuse, when interpreting the observed decline in antibiotic sales. In the revised version, we will expand the discussion to explicitly address these factors as possible influences alongside the e-Rx policy. (lines 465-472)

Minor revisions recommended and followed accordingly:

1.     The abstract should be segmented into “Background”, “Methods”, “Results”, and “Conclusions”. Here the “Background” sub-title is missing. – It is added. Thank you!

2.     Please consider adding “antibiotic stewardship” into the keywords. It is added. Thank you!

  1. Line 24, “The study applies” ---“applied”. Please keep the tense consistent in the abstract. If past tense is used in all other sentences, please use past tense here as well. -  Completely agree. Thank you. It is corrected.

4. Response to Comments on the Quality of English Language

Point 1: The English is fine and does not require any improvement.

Response 1: We sincerely thank the reviewer for the positive evaluation regarding the quality of the English language. We appreciate the acknowledgment and are pleased that the manuscript meets the expected linguistic standards. 

Round 2

Reviewer 2 Report (Previous Reviewer 3)

Comments and Suggestions for Authors

Most of the comments raised by the Reviewer are addressed.

One minor point left, the discrepancies in the pack numbers below must be corrected: 

Lines 234-236: The Access group experienced the largest absolute reduction − 74% (≈4.6 million packs), while Watch − 55% (≈2.5 million packs) and Reserve − 67% (≈0.9 million packs) also declined in parallel. 

Lines 243-244: Access −74% (≈4.6 million packs), Watch −55% (≈2.4 million packs), and Reserve −67% (≈0.8 million packs).

Author Response

Response to Reviewer 2 Comments

1. Summary

Thank you very much for taking the time to review our manuscript again. Please find the detailed responses and the corresponding revisions in the manuscript. In response to the reviewer’s recommendations, all the figures are in higher quality according to the journal’s requirements.

2. Questions for General Evaluation

Reviewer’s Evaluation

Response and Revisions

Does the introduction provide sufficient background and include all relevant references?

Yes/Can be improved/Must be improved/Not applicable

Thank you for the Reviewer’s evaluation!

Are all the cited references relevant to the research?

The references were carefully double-checked and were verified to be from reputable, peer-reviewed sources and consistently formatted according to the chosen citation style.

Is the research design appropriate?

Yes/Can be improved/Must be improved/Not applicable

Thank you!

Are the methods adequately described?

Yes/Can be improved/Must be improved/Not applicable

Thank you!

Are the results clearly presented?

Yes/Can be improved/Must be improved/Not applicable

Yes, the results are now clearly presented. The overall presentation of results has been improved accordingly. Thank you!

Are the conclusions supported by the results?

Yes/Can be improved/Must be improved/Not applicable

Thank you!

Are the figures clear and well-presented?

Yes/Can be improved/Must be improved/Not applicable

All the figures are improved according to the journal’s requirements - high quality, 600 dpi in PNG. Thank you for the recommendations, they encouraged us for improvement!

3. Point-by-point response to Comments and Suggestions for Authors

1)     Comments and Suggestions for Authors - 1:

One minor point left, the discrepancies in the pack numbers below must be corrected:

Lines 234-236: The Access group experienced the largest absolute reduction − 74% (≈4.6 million packs), while Watch − 55% (≈2.5 million packs) and Reserve − 67% (≈0.9 million packs) also declined in parallel.

Lines 243-244: Access −74% (≈4.6 million packs), Watch −55% (≈2.4 million packs), and Reserve −67% (≈0.8 million packs).

Answer – It is done. The numbers are corrected and are identical now. Thank you for your recommendation.

Reviewer 3 Report (New Reviewer)

Comments and Suggestions for Authors

The reviewer is glad to see that the authors have addressed the comments point-by-point, which improved the overall readability and significance of this manuscript.

The reviewer would like to point out that the the revisions corresponding to comment# 5 was not found in the revised manuscript. Please add the discussions about importance of considering additional potential confounders other than e-Rx.

Author Response

Response to Reviewer 3 Comments

1. Summary

Thank you very much for taking the time to review our manuscript again. Please find the detailed responses below and the corresponding revisions/corrections highlighted in the revised manuscript.

2. Questions for General Evaluation

Reviewer’s Evaluation

Response and Revisions

Does the introduction provide sufficient background and include all relevant references?

Yes/Can be improved/Must be improved/Not applicable

Thank you!

Are all the cited references relevant to the research?

The references were carefully double-checked and were verified to be from reputable. Additional References have been added and this is highlighted.

Is the research design appropriate?

Yes/Can be improved/Must be improved/Not applicable

The research design is aligned with the recommendations, and it is appropriate for the scope and aim of the study, as it combines policy analysis with real-world antibiotic sales data to assess the early effects of mandatory e-prescribing. We have fully addressed the reviewer’s previous recommendations, and this is reflected both in the revised manuscript.

Are the methods adequately described?

Yes/Can be improved/Must be improved/Not applicable

Thank you

Are the results and the figures and tables clearly presented?

Yes/Can be improved/Must be improved/Not applicable

Yes, Thank you! The results have been enriched accordingly, and the changes are highlighted in the text.

Are the conclusions supported by the results?

Yes/Can be improved/Must be improved/Not applicable

Yes, the conclusions are supported by the results. In line with the reviewer’s recommendation, we revisited the Discussion and Conclusion section and strengthened it accordingly.

3. Point-by-point response to Comments and Suggestions for Authors

Major comment - Comments:

    The reviewer is glad to see that the authors have addressed the comments point-by-point, which improved the overall readability and significance of this manuscript.

The reviewer would like to point out that the revisions corresponding to comment# 5 was not found in the revised manuscript. Please add the discussions about importance of considering additional potential confounders other than e-Rx

Response:

We sincerely thank the reviewer for this careful observation. It appears that the relevant text, although drafted and incorporated during revision, did not remain in the submitted version - likely due to a technical oversight. We fully agree with the reviewer on the importance of considering additional potential confounders when interpreting the observed decline in antibiotic sales. Accordingly, we have now explicitly added a discussion of other possible contributing factors, including public health interventions, disease incidence, supply chain disruptions, distribution practices, and inappropriate underuse, alongside the implementation of mandatory e-Rx. This discussion has been incorporated in the revised manuscript (≈lines 468–475).

This manuscript is a resubmission of an earlier submission. The following is a list of the peer review reports and author responses from that submission.

Round 1

Reviewer 1 Report

Comments and Suggestions for Authors

I appreciate the authors' effort to provide more accurate information on the evolution of antibiotic consumption in Bulgaria.

However, the lack of comparable information in the units considered as standard indicators represents a significant limitation for the correct interpretation of the results.

I believe it is essential to consider the introduction of these indicators to avoid biases resulting from their non-consideration.

Author Response

Response to Reviewer 1 Comments

1. Summary

Thank you very much for taking the time to review this manuscript. Please find the detailed responses below and the corresponding revisions/corrections highlighted in the re-submitted file.

2. Questions for General Evaluation

Reviewer’s Evaluation

Response and Revisions

Does the introduction provide sufficient background and include all relevant references?

Yes/Can be improved/Must be improved/Not applicable

The reviewer’s comments prompted us to revise the entire paragraph, which led to a clearer and more focused Introduction section. The aim is refined to focus more specifically on appropriate antibiotic use. Additionally, we have updated the title to better reflect the scope and aim of the study.

Are all the cited references relevant to the research?

The references were carefully double-checked and were verified to be from reputable, peer-reviewed sources and consistently formatted according to the chosen citation style.

Is the research design appropriate?

Yes/Can be improved/Must be improved/Not applicable

The study design aligns with the scope and aim, combining policy analysis with real-world sales data to assess the early effects of mandatory e-Rx. The use of the Health Policy Triangle and national IQVIA data provides a balanced view of implementation and outcomes. All reviewer recommendations have been addressed and are reflected in the revised manuscript and detailed responses.

Are the methods adequately described?

Yes/Can be improved/Must be improved/Not applicable

Yes, the methods are adequately described.

Are the results – the figures and tables clearly presented?

Yes/Can be improved/Must be improved/Not applicable

Yes, the results are clearly presented. Two figures have been added and some of the information has been moved to the Discussion part.

Are the conclusions supported by the results?

Yes/Can be improved/Must be improved/Not applicable

Yes, the conclusions are supported by the results. Following the reviewer’s recommendation, we refined the aim of the study, enriched the results with additional analyses, and expanded the discussion. As a result, the Conclusion section has been revised to more accurately reflect these updates and provide a clearer synthesis of the findings.

3. Point-by-point response to Comments and Suggestions for Authors

Comments 1:

I appreciate the authors' effort to provide more accurate information on the evolution of antibiotic consumption in Bulgaria.

However, the lack of comparable information in the units considered as standard indicators represents a significant limitation for the correct interpretation of the results.

I believe it is essential to consider the introduction of these indicators to avoid biases resulting from their non-consideration.

Response 1:

Thank you very much for your thoughtful and constructive feedback. We sincerely appreciate your recognition of our efforts to present a clearer overview of antibiotic consumption trends in Bulgaria. In response, we have revised all sections of the manuscript and refined the aim to focus more specifically on appropriate antibiotic use. Additionally, we have updated the title to better reflect the scope and aim of the study.

We fully agree that the absence of standardized indicators—such as Defined Daily Doses (DDD) per 1,000 inhabitants per day—represents a significant limitation, particularly when it comes to ensuring international comparability and avoiding potential interpretation biases. Due to current data availability constraints, we were not able to access DDD-based or population-adjusted indicators for this study. Nonetheless, we have now made this limitation more explicit in the revised manuscript.

To strengthen the analysis and provide a more nuanced picture, we have added additional figures:

-    Figure 2 -  presenting monthly antibiotic sales from 2022 to 2025, which allows for seasonal comparisons across years.

-        Figure 3 - showing annual sales disaggregated by AWaRe classification (Access, Watch, Reserve) to illustrate shifts in usage patterns among these key groups.

These additions aim to enhance the depth and transparency of the results, even in the absence of standard consumption metrics. We hope they contribute to a more informed understanding of the potential impacts of recent policy changes.

Once again, thank you for your valuable input, which has significantly contributed to the improvement of our work.

The changed text in the manuscript is highlighted.

4. Response to Comments on the Quality of English Language

Point 1: The English is fine and does not require any improvement.

Response 1: We sincerely thank the reviewer for the positive evaluation regarding the quality of the English language. We appreciate the acknowledgment and are pleased that the manuscript meets the expected linguistic standards.

Reviewer 2 Report

Comments and Suggestions for Authors

The manuscript describes the antibiotic consumption level at community level in Bulgaria over 3 years i.e. 2022-2024. The data has been presented as sales data acquired from IQVIA database, which is a standard and commonly used database for analysing sales data.

The authors mention the trends in antibiotic sales as proxy of consumption, prescribing and dispensing behaviours.  Though for consumption this proxy measure seems relevant; but in order to take sales data as proxy for prescribing and dispensing behaviours, we need to have some evidence regarding change in behaviour among actors i.e. prescribers, pharmacists as well as patients. Though authors have explained the roles of these actors and quoted few previous studies under “Results” section, but since they don’t seem to have themselves studied such behaviour change as part of this study, such text can be a part of discussion and not results.

Anti-bacterial sales data has been taken together as a group; were any efforts made to get data pertaining to trends in sales of different groups of antibiotics?

Lines no. 390-392: “According the IQVIA data, higher decline in sales (around 14%) encompasses classes 390 of antibiotics categorized under WHO’s AWaRe framework as Watch and Reserve, which 391 are associated with higher risks for resistance and are prioritized for stewardship..”,  how do the authors come to this conclusion when they have not presented the data in terms of different groups of antibiotics, particularly the AWaRe classes in this context.

Author Response

Response to Reviewer 2 Comments

1. Summary

Thank you very much for taking the time to review our manuscript. Please find the detailed responses in red below and the corresponding revisions highlighted in the manuscript.

2. Questions for General Evaluation

Reviewer’s Evaluation

Response and Revisions

Does the introduction provide sufficient background and include all relevant references?

Yes/Can be improved/Must be improved/Not applicable

Thank you for the Reviewer’s evaluation!

Are all the cited references relevant to the research?

The references were carefully double-checked and were verified to be from reputable, peer-reviewed sources and consistently formatted according to the chosen citation style.

Is the research design appropriate?

Yes/Can be improved/Must be improved/Not applicable

Thank you for the Reviewer’s evaluation!

Are the methods adequately described?

Yes/Can be improved/Must be improved/Not applicable

Yes, the methods are adequately described. Following the reviewer’s suggestions, we have clarified specific aspects of the methodology in the revised manuscript. As a result of the reviewer’s comments the title of the article and the aim were made more focused. Thank you!

Are the results clearly presented?

Yes/Can be improved/Must be improved/Not applicable

Yes, the results are clearly presented. Following the reviewer’s valuable comment some clarifications are now included in the manuscript, and the overall presentation of results has been improved accordingly, as highlighted in the manuscript. Two additional figures have been added and some of the analyses have been moved to discussion as proposed by the reviewer. Thank you!

Are the conclusions supported by the results?

Yes/Can be improved/Must be improved/Not applicable

With the enriched results and the revised discussion, we believe the conclusions are now well supported by the presented data.

Are the figures clear and well-presented?

Yes/Can be improved/Must be improved/Not applicable

Thank you for the assessment.

3. Point-by-point response to Comments and Suggestions for Authors

1)      Comments and Suggestions for Authors - 1:

Comment 1: The manuscript describes the antibiotic consumption level at community level in Bulgaria over 3 years i.e. 2022-2024. The data has been presented as sales data acquired from IQVIA database, which is a standard and commonly used database for analysing sales data.

Response 1:  

We sincerely thank the reviewer for this supportive and thoughtful comment. We are grateful for your careful reading and for acknowledging the appropriateness of our methodology, particularly the use of the IQVIA database as a standard and widely accepted source for analyzing antibiotic sales data. As noted in the manuscript, IQVIA Bulgaria kindly provided this reliable data as part of their commitment to social responsibility and support for academic research.

2 (a) - Comments and Suggestions for Authors:

The authors mention the trends in antibiotic sales as proxy of consumption, prescribing and dispensing behaviours. Though for consumption this proxy measure seems relevant; but in order to take sales data as proxy for prescribing and dispensing behaviours, we need to have some evidence regarding change in behaviour among actors i.e. prescribers, pharmacists as well as patients.

Response 2 (a):

We thank the reviewer for this insightful and well-founded comment. We fully agree that while sales data can serve as a reasonable proxy for trends in antibiotic consumption, it does not, in itself, provide sufficient evidence to draw conclusions about prescribing and dispensing behaviours without additional data on behavioural changes among key actors - namely prescribers and pharmacists. In response, we have revised the manuscript title and refined the aim to focus more precisely on trends in antibiotic consumption.

The changes are highlighted in the manuscript and in the abstract (Lines 63-64; 19-22).

    2 (b) Comments and Suggestions for Authors:

Though authors have explained the roles of these actors and quoted few previous studies under “Results” section, but since they don’t seem to have themselves studied such behaviour change as part of this study, such text can be a part of discussion and not results.

Response 2 (b): – Fully agree, it is done (see lines 341-371). Thank you!

3 (a) - Comments and Suggestions for Authors:

Anti-bacterial sales data has been taken together as a group; were any efforts made to get data pertaining to trends in sales of different groups of antibiotics?

Response 3 (a): We thank the reviewer for this thoughtful and constructive comment. In response, we have further explored the sales trends by disaggregating the data according to the AWaRe classification of antibiotics. This analysis is now presented in a new figure (see Figure 3, lines 299–311), which illustrates the annual sales of antibiotics in community settings by AWaRe group for the period 2022–2024. We hope this addition provides greater clarity and value to the manuscript. The analyses are included in the discussion section accordingly (lines 449-463).

3 (b) - Comments and Suggestions for Authors:

Lines no. 390-392: “According the IQVIA data, higher decline in sales (around 14%) encompasses classes of antibiotics categorized under WHO’s AWaRe framework as Watch and Reserve, which are associated with higher risks for resistance and are prioritized for stewardship.”,  how do the authors come to this conclusion when they have not presented the data in terms of different groups of antibiotics, particularly the AWaRe classes in this context.

Response 3 (b): Thank you for this important observation. As noted in our response above, we have now included a dedicated figure presenting antibiotic sales by AWaRe groups (see Figure 3, lines 299–311). This addition provides the necessary data to support the conclusion regarding the decline in sales of Watch and Reserve antibiotics.

4. Response to Comments on the Quality of English Language

Point 1: The English is fine and does not require any improvement.

Response 1: We sincerely thank the reviewer for the positive evaluation regarding the quality of the English language. We appreciate the acknowledgment and are pleased that the manuscript meets the expected linguistic standards.

Reviewer 3 Report

Comments and Suggestions for Authors

This article summarizes authors’ analyses on the effect of implementation of mandatory electronic prescription of antibiotics on the antimicrobial usage in Bulgaria. As the importance of finding effective antimicrobial therapies and antimicrobial stewardship approaches rises day by day, these findings become more valuable for increasing number of researchers. However, there are several major issues that need to be addressed before the manuscript can be accepted for publication.

  1. Section 3.4. is titled as “Sales data on antibiotics at a community level – 2019–2025”, but the data presented in the section only refers to the years 2022, 2023 and 2024.
  2. Major conclusion drawn by the authors in the manuscript is the claim that implementation of mandatory electronic prescription and thus documentation of the corresponding prescriptions reduced the overuse of antibiotics in Bulgaria. However, according to data in Figure 2, monthly sales of antibiotics decrease with the same pace during the first half of year 2023 (months before the introduction of electronic prescription in October 2023 and one year before the full implementation of electronic prescription in April 2024) and second half of the year 2024. Thus, the main conclusion presented in this study has no roots in the data presented.  
  3. How do the changes in seasonal use of antimicrobials correlate with the data presented in Figure 2? Monthly antimicrobial use of the country must be presented for a longer period. Moreover, the data must be corrected for seasonal demand for antimicrobials by comparing the sales data for each month/quarter with the data from the same time period of the previous years.  

Author Response

Response to Reviewer 3 Comments

1. Summary

Thank you very much for taking the time to review this manuscript. Please find the detailed responses below and the corresponding revisions/corrections highlighted in the re-submitted file.

2. Questions for General Evaluation

Reviewer’s Evaluation

Response and Revisions

Does the introduction provide sufficient background and include all relevant references?

Yes/Can be improved/Must be improved/Not applicable

The introduction was revised, which led to a clearer section and more focused aim. These changes are highlighted in the manuscript and detailed in the point-by-point response below.

Are all the cited references relevant to the research?

The references were carefully double-checked and were verified to be from reputable. Additional References have been added and this is highlighted.

Is the research design appropriate?

Yes/Can be improved/Must be improved/Not applicable

The research design is aligned with the recommendations and it is appropriate for the scope and aim of the study, as it combines policy analysis with real-world antibiotic sales data to assess the early effects of mandatory e-prescribing. The use of the Health Policy Triangle framework, alongside quantitative trends from a reliable national data source (IQVIA Bulgaria), allows further analyses.

We have fully addressed the reviewer’s recommendations, and this is reflected both in the revised manuscript and in our step-by-step responses below.

Are the methods adequately described?

Yes/Can be improved/Must be improved/Not applicable

Yes, the methods are adequately described. All the reviewer’s suggestions are accepted and applied.

Are the results – the figures and tables clearly presented?

Yes/Can be improved/Must be improved/Not applicable

Yes, the results are clearly presented. Additional figures have been made, and the result section is optimized. 

Are the conclusions supported by the results?

Yes/Can be improved/Must be improved/Not applicable

Yes, the conclusions are supported by the results. In line with the reviewer’s recommendation, we revisited the Conclusion section and strengthened it accordingly.

3. Point-by-point response to Comments and Suggestions for Authors

Comments 1:

This article summarizes authors’ analyses on the effect of implementation of mandatory electronic prescription of antibiotics on the antimicrobial usage in Bulgaria. As the importance of finding effective antimicrobial therapies and antimicrobial stewardship approaches rises day by day, these findings become more valuable for increasing number of researchers. However, there are several major issues that need to be addressed before the manuscript can be accepted for publication.

Response 1: Thank you very much for your thoughtful and constructive comments. We truly appreciate your recognition of the importance of our work and your helpful feedback. We carefully addressed all the issues raised to improve the manuscript and ensure it meets the highest standards.

The changed text in the manuscript is highlighted.

Comments 2: Regarding the figure 2

    Major conclusion drawn by the authors in the manuscript is the claim that implementation of mandatory electronic prescription and thus documentation of the corresponding prescriptions reduced the overuse of antibiotics in Bulgaria. However, according to data in Figure 2, monthly sales of antibiotics decrease with the same pace during the first half of year 2023 (months before the introduction of electronic prescription in October 2023 and one year before the full implementation of electronic prescription in April 2024) and second half of the year 2024. Thus, the main conclusion presented in this study has no roots in the data presented. 

How do the changes in seasonal use of antimicrobials correlate with the data presented in Figure 2? Monthly antimicrobial use of the country must be presented for a longer period. Moreover, the data must be corrected for seasonal demand for antimicrobials by comparing the sales data for each month/quarter with the data from the same time period of the previous years. 

Respons 2:

Thank you for this valuable and constructive comment. We appreciate your close reading of the data and the opportunity to clarify and strengthen our conclusions.

In response, we have expanded our analysis and included additional figures to better account for seasonality and inter-annual trends:

    Figure 2 (new) presents monthly antibiotic sales for 2022–2025, enabling direct comparison between corresponding months across years. This allows for a clearer understanding of seasonal patterns and changes in antimicrobial use.

    Figure 3 (new) shows annual antibiotic sales by AWaRe category, highlighting significant reductions—particularly in the Watch and Reserve groups—between 2023 and 2024, which are key indicators for antimicrobial stewardship.

    Figure 4 (previously Figure 2) is now contextualized within this broader dataset. While a general downward trend is evident throughout 2023, we observe a plateau during the e-prescription moratorium (October 2023 – March 2024), followed by a sharper decline post-enforcement in April 2024. This temporal association strengthens the hypothesis of an effect related to policy implementation.

Additionally, we have revised the title and focused the aim of the article to more accurately reflect the scope of our analysis—namely, examining trends in antibiotic sales in the context of the gradual implementation of electronic prescriptions, while explicitly acknowledging that this is a temporal correlation rather than a definitive causal relationship.

We have also adjusted the conclusions accordingly to ensure they are data-driven and cautious in interpretation. We are grateful for your suggestions, which have led to significant improvements in the manuscript’s clarity and analytical rigor.

The changed text and added figures in the manuscript are highlighted.

4. Response to Comments on the Quality of English Language

Point 1: The English is fine and does not require any improvement.

Response 1: We sincerely thank the reviewer for the positive evaluation regarding the quality of the English language. We appreciate the acknowledgment and are pleased that the manuscript meets the expected linguistic standards.